# Marker-Free Rice (*Oryza sativa* L. cv. IR 64) Overexpressing *PDH45* Gene Confers Salinity Tolerance by Maintaining Photosynthesis and Antioxidant Machinery

**DOI:** 10.3390/antiox11040770

**Published:** 2022-04-12

**Authors:** Ranjan Kumar Sahoo, Renu Tuteja, Ritu Gill, Juan Francisco Jiménez Bremont, Sarvajeet Singh Gill, Narendra Tuteja

**Affiliations:** 1Department of Biotechnology, School of Engineering and Technology, Centurion University of Technology and Management, Bhubaneswar 752050, Odisha, India; ranjan.sahoo@cutm.ac.in; 2International Centre for Genetic Engineering and Biotechnology, Aruna Asaf Ali Marg, New Delhi 110067, India; renu@icgeb.res.in; 3Stress Physiology & Molecular Biology Lab, Centre for Biotechnology, Maharshi Dayanand University, Rohtak 124001, Haryana, India; ritu_gill@mdurohtak.ac.in; 4Laboratorio de Biotecnología Molecular Plantas, Division de Biología Molecular, Instituto Potosino de Investigacion Científicay Tecnologica AC, San Luis Potosí 78395, Mexico

**Keywords:** antioxidants, reactive oxygen species, oxidative stress, marker-free transgenic rice, mature seed-derived calli, pea DNA helicase 45, photosynthesis, salinity stress tolerance

## Abstract

Helicases function as key enzymes in salinity stress tolerance, and the role and function of *PDH45* (pea DNA helicase 45) in stress tolerance have been reported in different crops with selectable markers, raising public and regulatory concerns. In the present study, we developed five lines of marker-free *PDH45*-overexpressing transgenic lines of rice (*Oryza sativa* L. cv. IR64). The overexpression of *PDH45* driven by CaMV35S promoter in transgenic rice conferred high salinity (200 mM NaCl) tolerance in the T_1_ generation. Molecular attributes such as PCR, RT-PCR, and Southern and Western blot analyses confirmed stable integration and expression of the *PDH45* gene in the *PDH45*-overexpressing lines. We observed higher endogenous levels of sugars (glucose and fructose) and hormones (GA, zeatin, and IAA) in the transgenic lines in comparison to control plants (empty vector (VC) and wild type (WT)) under salt treatments. Furthermore, photosynthetic characteristics such as net photosynthetic rate (Pn), stomatal conductance (gs), intercellular CO_2_ (Ci), and chlorophyll (Chl) content were significantly higher in transgenic lines under salinity stress as compared to control plants. However, the maximum primary photochemical efficiency of PSII, as an estimated from variable to maximum chlorophyll a fluorescence (Fv/Fm), was identical in the transgenics to that in the control plants. The activities of antioxidant enzymes, such as catalase (CAT), ascorbate peroxidase (APX), glutathione reductase (GR), and guaiacol peroxidase (GPX), were significantly higher in transgenic lines in comparison to control plants, which helped in keeping the oxidative stress burden (MDA and H_2_O_2_) lesser on transgenic lines, thus protecting the growth and photosynthetic efficiency of the plants. Overall, the present research reports the development of marker-free *PDH45*-overexpressing transgenic lines for salt tolerance that can potentially avoid public and biosafety concerns and facilitate the commercialization of genetically engineered crop plants.

## 1. Introduction

Rice (*Oryza sativa* L., family Gramineae (Poaceae)) is an important staple food crop that is produced (518 million tonnes, milled), cultivated, and consumed globally in >122 countries (excluding Antarctica), being susceptible to salt amongst cereal crops [1,2,3]. Major abiotic stresses (salinity, drought, extreme temperatures, heavy metal, etc.) are a significant limitation in rice cultivation globally [1]. Soil salinity is a major problem that reduces productivity of crops in irrigated as well as in tropical fields, where the deterioration of agricultural lands occur due to salinity [4,5,6]. It brings series of changes at the physiological, biochemical, and molecular levels by affecting the photosynthetic machinery (partial stomatal closure and hampered photosystem II (PSII), reactive oxygen species (ROS)-led molecular injury, restricted water/nutrient availability, and disturbed sodium (Na^+^)/potassium ion (K^+^) homeostasis), which ultimately poses serious yield penalty [7,8,9,10,11,12]. Due to rapidly growing global population and urbanization, it is impossible to increase the cultivated land area, and therefore to fulfill the demand of rice consumers, it becomes imperative to discover new techniques for developing salinity-tolerant crop plants by protecting the photosynthetic machinery (net photosynthetic rate, stomatal conductance, chlorophyll content), efficient ROS scavenging, membrane integrity, Na^+^ exclusion, etc. [13,14,15]. Robust antioxidant machinery consisting of enzymatic (SOD, CAT, APX, GPX, GR, etc.) and non-enzymatic antioxidants (glutathione (GSH) and ascorbic acid (AsA) is efficient enough to protect the photosynthetic machinery, cellular components, and membranes under various abiotic stresses [8]. Therefore, strong antioxidant machinery can be well correlated with salinity stress tolerance in crop plants [8]. Nidumukkala et al. [16] reviewed the fact that overexpression of helicases in different model and crop plants provides salt tolerance though increased antioxidant capacity, photosynthetic efficiency, and ion homeostasis, as well as by regulating the expression of various stress responsive genes. Therefore, introduction of a stress-tolerant gene in rice is one of the effective ways to develop stress-tolerant cultivars without yield penalty. The presence of selectable marker genes (SMGs, antibiotic or herbicide resistance genes) in genetically engineered crops may arouse public and regulatory concerns due to biosafety issues because the weeds or pathogenic microorganisms present in soil may become resistant to herbicides or antibiotics and can harm public health [17]. The problem of transgene expression arises due to the sexual crossing, which can lead to homology-dependent gene silencing in the genome [17]. Due to consumer, environmental, and biosafety concerns, the regulatory bodies also encourage the development of marker-free transgenic crops with an array of different transformation strategies such as homologous recombination, site-specific recombination, co-transformation, transposon-mediated transgene reintegration system, and CRISPR/Cas9 system [17,18,19,20]. The tissue culture methods are generally used to understand the mechanisms underlying salt tolerance of transgenic lines [21,22]. Several techniques have been developed to improve Agrobacterium-mediated transformation of indica rice [23,24,25]. The development of an efficient large-scale transformation system requires a large number of transformants for successful gene transfer [24]. Previously, many researchers developed a transformation protocol for marker-free transgenic rice plants using anther culture [26,27], but the unavailability of explants (anther) throughout the year is a major limitation of this method and it is very laborious to screen the transgenic plants by a PCR-based method.

In the present study, we report that overexpression of *PDH45* gene in an elite indica rice variety IR64 (*Oryza sativa* L., cv. IR64) showed tolerance against salinity stress as well as improved growth, photosynthesis, and better antioxidant machinery in the transgenic rice. We exploited the potential of transgenic technologies for crop improvement through developing marker-free transgenic *PDH45* rice. Thus, we also successfully developed a screening technique using 200 mM NaCl salt to screen marker-free *PDH45* transgenic rice plants. Development of rice transgenic lines overexpressing the *PDH45* gene without the antibiotic marker gene for stable expression of the stress-tolerant trait in a predictable manner avoids the transfer of undesirable transgenic material to non-transgenic crops and related species.

## 2. Materials and Methods

### 2.1. Cloning and Transformation of PDH45 Gene in IR64 Rice

*PDH45* gene (accession number: Y17186) was used to establish the tissue culture technique. The coding region of *PDH45* gene (1.2 kb) was cloned in reporter gene-free plant transformation vector pCAMBIA1300 in place of hygromycin to generate complete reporter and antibiotic marker-free plasmid pCAMBIA1300-*PDH45*. An empty vector (pCAMBIA1300) construct, called vector control (VC), was used to compare the function of the gene, and the VC construct comprises all components except the *PDH45* gene. The above two constructs (pCAMBIA1300-*PDH45* and pCAMBIA1300) were used for the *Agrobacterium tumefaciens* (LBA4404)-mediated transformation method [28]. The same conditions were used to generate all the plants.

### 2.2. Development of Selection Technique for Marker-Free Transgenic Plants

A new selection technique was developed by adding 200 mM NaCl in selection media, shoot induction, and root induction media for the selection of *PDH45* marker-free transgenic plants during the plant induction stage. We modified the media described by Sahoo and Tuteja [28]. Here, we used 200 mM NaCl in place of hygromycin as the gene *PDH45* has already been reported as being responsible for salinity tolerance in different plants [29,30,31,32,33,34]. The other compositions of media were the same as described earlier [28].

### 2.3. Molecular Analysis (PCR, Southern Blot, qRT-PCR, and Western Blot) of T1 Transgenic PDH45 Plants

The genomic DNA was extracted from the healthy leaves of marker-free *PDH45* transgenic plants and used to check the integration of the gene by PCR and Southern blot analysis. About 25 µg of genomic DNA was used for Southern blot analysis. First, the genomic DNA was digested with XbaI and resolved on 0.8% agarose gel followed by transfer to nylon membrane (Hybond-N, Amersham, Inc., Amersham, UK) as previously described [35]. The probe was radiolabelled by the gene amplification method using α–[32P] dCTP. Hybridization with the probe was conducted using the method described [35]. The qRT-PCR experiment was performed to check the transcript levels of the gene using gene-specific primers such as forward 5′-ATGGCGACAACTTCTGTG-3′ and reverse 5′-TATATAAGATCACCAATATTCATTGG-3′. For Western blot analysis, the crude plant extract was denatured and separated by SDS PAGE and transferred onto a polyvinylidene fluoride (PVDF) membrane using the method described [36]. Polyclonal antibodies (1:1000 dilutions) from rabbit were used as a probe against the *PDH45* gene.

### 2.4. Leaf Disk Senescence Assay and Chlorophyll Content

The chlorophyll content after leaf disk senescence assay was measured using the method described earlier [37].

### 2.5. Biochemical Analysis of Antioxidant Activities of Marker-Free PDH45 Transgenic Lines

The seeds of *PDH45* transgenic, WT, and VC plants were kept in hydroponics for germination, and 21-d-old plants were dipped in 200 mM NaCl for 24 h. The experiments were conducted in green houses of the International Centre for Genetic Engineering and Biotechnology (ICGEB), New Delhi, where 16 h light photoperiod at 25 °C temperature was maintained. Similar stress treatment and stress conditions as described were also used in the present study [12]. After 24 h salt stress, the plant tissues were used for biochemical analysis such as catalase (CAT), ascorbate peroxidase (APX), glutathione reductase (GR), proline, hydrogen peroxide (H_2_O_2_), lipid peroxidation, relative water content (RWC), and electrolytic leakage. All the parameters were measured using the methods described earlier [38].

### 2.6. Measurement of Photosynthetic Activities and Agronomic Characteristics of PDH45 Transgenic Plants

The different photosynthetic measurements such as photosynthetic yield, rate, intercellular CO_2_ concentration, CO_2_ release, stomatal conductance, and transpiration rate were recorded using an infrared gas analyzer (IRGA; LI-COR, http://www.licor.com, (accessed on 2 November 2021), on a sunny day between 11:00 and 12:00 noon. The plants were grown under 200 mM NaCl stress in a large tank, and all the parameters were measured using the expanded leaves of mature plants (60 d old). After 12 d of salt stress, different agronomic characteristics were measured using the method described earlier [12].

### 2.7. Chlorophyll a Fluorescence Measurements

Plants were grown in green houses of the International Centre for Genetic Engineering and Biotechnology (ICGEB), New Delhi, where 16 h light [photosynthetically active radiation (750 µmol m^−2^ s^−1^)] photoperiod at 25 °C temperature was maintained. Minimal fluorescence (Fo), maximal fluorescence (Fm), maximal variable fluorescence (Fv), and Fv/Fm ratio were included, where Fv = Fm − Fo.

Chlorophyll a (Chl a) fluorescence from the leaves of 25-day-old WT, VC, and transgenic rice seedlings was measured with a PAM-2100 fluorometer (Walz, Germany). Before each measurement, the leaf sample was kept in the dark for 20 min [39]. Optimum quantum efficiency (uPSII, also referred to as Y) of Photosystem II (PSII) was inferred from Fv/Fm = (Fm − Fo)/Fm [40].

### 2.8. Estimation of Sugar, Hormones (GA, Zeatin and IAA), and Ion Contents

Shoots and roots from mature (60 d old) *PDH45* T1 transgenic, VC, and WT plants after 12 d of salt stress were used in this study. The sugar content was estimated as described earlier [41]. The endogenous plant hormones (GA, zeatin and IAA) were estimated as described earlier [42]. The flame ionization photometer was used for the estimation of potassium, as described by Chapman and Pratt [43]. The sodium content was estimated as described by Munns et al. [44].

### 2.9. Salinity Tolerance of Transgenic Plant under 200 mM NaCl Stress

The *PDH45* transgenic lines (L4, L7, L8, L11 and L13) and VC and WT rice plants (60 d old) were grown in one large metal pot filled with soil and dipped in 200 mM NaCl. The plants were allowed to grow up to maturity (harvest), and the phenotypic conditions of these plants were recorded.

### 2.10. Statistical Analysis

The experimental data were statistically analyzed, and standard error was calculated from three independent observations. Analysis of variance (ANOVA) was performed on the data using SPSS (10.0 Inc., Chicago, IL, USA) to determine the least significant difference (LSD) for the significant data to identify the differences between means and presented as mean ± SE. The means were separated by Duncan’s multiple range tests. Different letters indicate significant difference at *p* < 0.05.

## 3. Results

### 3.1. Molecular Analysis of Marker-Free PDH45 Transgenic Lines

The marker-free *PDH45* transgenic IR64 rice plants were developed using the pCAMBIA1300-*PDH45* gene construct (Figure 1a). Phenotypically, the transgenic rice plants were not significantly different from WT and VC plants (Figure 1b). The desired *PDH45* gene (1.2 kb) fragment was detected by PCR (Figure 1c). The Southern blot results confirmed the integration of a single-copy *PDH45* gene in transgenic rice plants in all the five transgenic lines (L4, L7, L8, L11, and L13) (Figure 1d). The real-time PCR (qRT-PCR) provided ≈8-fold induction in the transcript level of *PDH45* in transgenic lines (L4, L7, L8, L11 and L13) (Figure 1e). The Western blot results showed that *PDH45* protein was expressed to almost similar levels in all the transgenic lines (L4, L7, L8, L11 and L13) as compared to WT and VC plants (Figure 1f).

### 3.2. PDH45 Transgenic Lines Showed Salinity Tolerance

The damage caused in the leaf pieces by salt stress was observed in all the plants after 72 h; however, the *PDH45*-overexpressing lines displayed darker green leaves, in contrast to the yellowish leaves of the WT and VC plants (Figure 2a). In this sense, the reduction of chlorophyll content in leaf tissues was lesser in transgenic lines as compared to WT and VC plants under salt stress (Figure 2b). The lesser chlorophyll content in the leaf tissues of WT and VC plants as compared to transgenic lines provided strong evidence towards tolerance against salinity stress (Figure 2a,b). The transgenic lines (L4, L7, L8, L11 and L13) along with WT and VC plants were allowed to grow up to maturity in a metal tank filled with 200 mM NaCl. After 3d, WT and VC plants showed dropping characteristics, whereas *PDH45*-overexpressing transgenic lines L4, L7, L8, L11 and L13 grew well and produced viable seeds (Figure 2c,d).

### 3.3. Agronomic Performance of Marker-Free PDH45 Transgenic Plants under Stress

The agronomic performance of T_1_ transgenic lines under 200 mM NaCl treatment was compared with WT and VC without NaCl treatment. Better agronomic characteristics were observed in *PDH45* transgenic plants as compared to WT and VC plants. Several phenotypic characteristics of transgenic plants were recorded and found to be almost similar to the WT and VC plants grown in 0 mM NaCl. However, under 200 mM NaCl treatment, the WT and VC plants did not survive until flowering stage (Figure 2d).

### 3.4. Photosynthetic Characteristics and Endogenous Ion Content of Marker-Free PDH45 T1 Transgenic Plants under Stress

The photosynthetic characteristics of transgenic plants were observed and compared to WT and VC plants after 12 d of induction of 200 mM NaCl salt treatment. The photosynthetic rate declined by 33% in WT and 35% in VC plants as compared to *PDH45* marker-free transgenic lines. The net photosynthetic rate, stomatal conductance, intracellular CO_2_, CO_2_ release, and transpiration rate were also higher in transgenic lines as compared to the WT and VC plants (Figure 3a–e).

### 3.5. Chlorophyll a Fluorescence

The chlorophyll fluorescence rose from a low minimum level (‘‘O’’ level or Fo) to a higher maximum level (‘‘P’’ level or Fm) when exposed from dark to light. The maximum primary photochemical efficiency of PSII, estimated from Fv/Fm, was almost identical in the transgenics to that in the VC and WT (Figure 3f).

### 3.6. Analysis of MDA, H_2_O_2_, Ion Leakage, and Antioxidant Response in Marker-Free PDH45 T1 Transgenic Plants

The salt-induced changes in the ion leakage, H_2_O_2_, proline content, accumulation of MDA, RWC, and antioxidant machineries in T_1_
*PDH45* transgenic lines (L4, L7, L8, L11 and L13) were compared with WT and VC rice seedlings. We observed reduced levels of MDA, H_2_O_2_, and ion leakage, alongside an increase in proline content in *PDH45* transgenic lines in comparison to the WT and VC plants under salt stress at 200 mM NaCl (Figure 4a–d). The activities of CAT, APX, GPX, GR and RWC were increased in *PDH45* transgenic plants as compared to WT and VC plants (Figure 4e–i).

### 3.7. The Sugar and Hormone Content of Marker-Free PDH45 T1 Transgenic Plants

The *PDH45* L4, L7, L8, L11 and L13 transgenic lines showed higher endogenous sugar (glucose and fructose) content in roots as well as in shoots when compared with WT and VC plants (Figure 5a,b). The endogenous hormones such as GA, zeatin, and IAA content were also higher in roots and shoots of *PDH45* transgenics as compared to WT and VC plants (Figure 5c–e). The potassium content in transgenic plants was higher, whereas sodium content was lower in marker-free *PDH45* transgenic plant tissues as compared to WT and VC plants (Figure 5f).

## 4. Discussion

In the era of frequently changing global climatic conditions, shortage of irrigation water, reduced agriculturally suitable cultivable land area, degradation and salinization of the agricultural soil, and unpredictable onset of abiotic stresses, agricultural productivity is severely affected, posing a serious threat to food security. Therefore, it is imperative to develop genetically engineered stress-tolerant crops with all the qualifications of global acceptance. It has been reported that overexpression of helicases (*PDH45*/*PDH47*) in different model and crop plants provides salt/cold tolerance through increased antioxidant capacity, photosynthetic efficiency, and ion homeostasis, as well as by regulating the expression of various stress responsive genes [16,29,30,31,32,37,45]. Genetically engineered transgenic crops with selectable markers (antibiotic or herbicide resistance) have public and regulatory concerns; therefore, development of marker-free transgenic plants is needed in order to avoid public and biosafety concerns and to facilitate the commercialization of genetically engineered crop plants [17].

We developed the method to select marker and reporter free transgenic lines using the previously published reports [46,47]. In this research, marker-free *PDH45* transgenic rice plants were raised using Agrobacterium-mediated transformation followed by screening with 200 mM NaCl in selection, shoot, and root regeneration media to select only the transformed calli for plant regeneration because *PDH45* is responsible for salinity tolerance [31,32,33,45]. The elevated stress tolerance in *PDH45*-expressing plants correlated with MH1 (*M*. *sativa* helicase 1) transgenic plants, showing that MH1 functions in abiotic stress tolerance by elevating reactive oxygen species (ROS) burden and through osmotic adjustment [48]. Five independent transgenic lines (L4, L7, L8, L11 and L13) along with empty VC and WT plants were used for functional validation under salt stress. These lines express almost similar levels of *PDH45* protein. Similar to previous reports, these *PDH45* transgenic rice plants also showed high salinity tolerance. This was indicated by the presence of higher chlorophyll content in the leaf disks of salinity-stressed T_1_ transgenic plants, whereas VC and WT plant leaves became yellow. Moreover, the transgenic plants were able to grow in the continuous presence of 200 mM NaCl stress. These results indicate that the introduced trait is functional in transgenic plants and that it is also stable. The transgenic lines also maintained higher endogenous nutrient contents as compared to the VC and WT plants under salinity stress, which revealed the salinity tolerance potential of the transgenic lines. Similar findings have been reported earlier [12,32,33,45,49]. Higher concentration of potassium and lower concentration of sodium were found in leaves of *PDH45*-overexpressing transgenic lines as compared with VC and WT plants.

*PDH45*-overexpressing marker-free transgenic lines maintained higher endogenous nutrient contents under salinity stress as compared with WT and VC plants, which proved the salt stress tolerance potential of the marker-free *PDH45* transgenic lines, which is in agreement with the previous reports [31,32,33]. The higher potassium and lower sodium concentration in T_1_ transgenic plants indicates that the lower Na+/K+ ratio in the transgenic lines might be responsible for imparting better stress tolerance to salinity stress in comparison to the VC and WT plants. The better photosynthetic activities such as net photosynthetic rate, stomatal conductance, intercellular CO_2_ concentration, CO_2_ release and transpiration rate, and photosynthetic yield (Fv/Fm) were observed in *PDH45* transgenic lines as compared to the VC and WT plants. The retention of chlorophyll content in transgenic lines indicates the better control over the photosynthetic apparatus under salt stress. Our data are in agreement with the earlier reports on *PDH45-*, SUV3-, and BAT1-overexpressing rice plants under stress [12,31,32,49].

Sugars such as glucose and fructose may play a key role in salt defense mechanisms through ROS detoxification [49,50,51,52,53]. The sugar content in *PDH45*-overexpressing marker-free transgenic lines was higher as compared to VC and WT plants. The *PDH45*-overexpressing transgenic rice plants showed significantly higher endogenous content of plant hormones in leaf, stem, and root, directing the molecular and biochemical mechanisms to confer increased stress tolerance [54]. A similar trend of endogenous plant hormone profile was also reflected in OsSUV3 and OsBAT1 transgenic rice under stress conditions [7,49]. This is a very simple, reproducible, and improved protocol for selection of marker-free transgenic rice plants using Agrobacterium-mediated transformation of mature seed-derived callus tissues of indica rice variety, IR64.

## 5. Conclusions

The present study provides reporter and marker-free transgenic rice plants that has a scope for future commercialization and approval from regulatory agencies as they are focusing on the removal of reporter and marker genes from transgenic plants. We developed a unique successful salt screening method for screening transgenic lines during tissue culture and also utilized the unique function of *PDH45* helicase in providing salt tolerance in marker-free transgenic rice cv. IR64. It also provides a good example for the exploitation of helicases for enhanced agricultural production, while withstanding extreme climatic conditions, maintaining biosafety regulations, and ensuring food security.

## Figures and Tables

**Figure 1 antioxidants-11-00770-f001:**
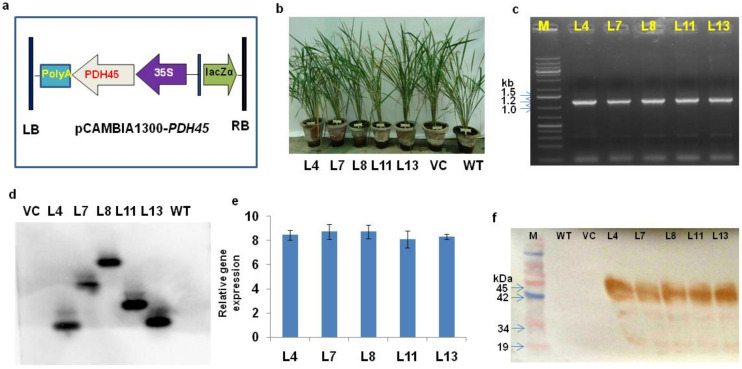
Screening and analysis of *PDH45* marker-free transgenic lines. (**a**) T-DNA construct of pCAMBIA 1300-*PDH45.* (**b**) Transgenic lines (L4, L7, L8, L11, L13, VC, and WT). (**c**) PCR conformation of the *PDH45*-overexpressing transgenic (T_1_) lines showed the amplification of 1.2 Kb fragment. (**d**) Southern blot analysis showing the integration and copy number of the *PDH45* gene. (**e**) Relative gene expression of *PDH45* transgenic lines. (**f**) Western blot analysis showing the *PDH45* protein (≈45 kDa).

**Figure 2 antioxidants-11-00770-f002:**
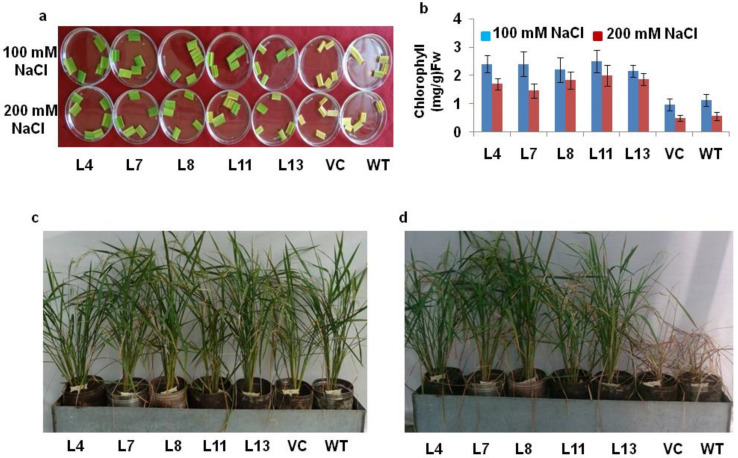
Salinity tolerance of *PDH45*-overexpressing transgenic T_1_ IR64 rice lines. (**a**) Leaf disk senescence assay under 100 and 200 mM NaCl treatment. (**b**) Chlorophyll content (mg/g fw) in *PDH45* transgenic lines after salt treatment. (**c**) Third day in 200 mM NaCl treatment. (**d**) After 15 days of NaCl treatment.

**Figure 3 antioxidants-11-00770-f003:**
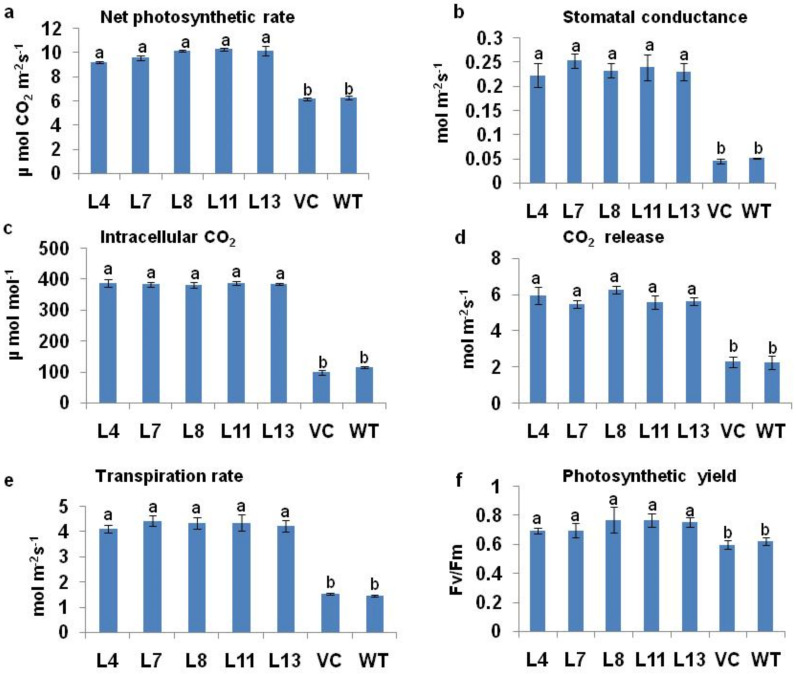
Measurement of photosynthetic characteristics and chlorophyll a fluorescence of WT, VC, and *PDH45* marker-free transgenic lines (L4, L7, L8, L11, and L13) under 200 mM NaCl treatment. (**a**) Photosynthetic rate. (**b**) Stomatal conductance. (**c**) Intracellular CO_2_. (**d**) CO_2_ release. (**e**) Transpiration rate. (**f**) Photosynthetic yield (Fv/Fm). Values are mean ± SE (*n* = 3). Different letters on the top of bars indicate significant differences at *p* ≤ 0.05 level as determined by Duncan’s multiple range test (DMRT).

**Figure 4 antioxidants-11-00770-f004:**
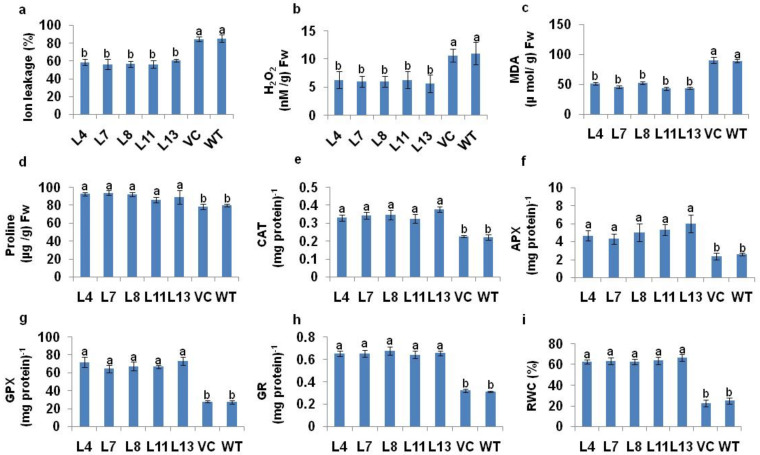
Biochemical analysis of *PDH45*-overexpressing T_1_ transgenic lines (L4, L7, L8, L11, L13, VC) and WT plants exposed to 24 h at 200 mM NaCl treatment. (**a**) Ion leakage. (**b**) Hydrogen peroxide (H_2_O_2_) content. (**c**) Lipid peroxidation expressed in terms of MDA content. (**d**) Level of proline accumulation. (**e**) Catalase (CAT) activity; one unit of enzyme activity defined as 1 μmol H_2_O_2_ oxidized min^−1^. (**f**) Ascorbate peroxidase (APX) activity; one unit of enzyme activity defined as 1 μmol of ascorbate oxidized min^−1^. (**g**) Guaiacol peroxidase (GPX) activity. (**h**) Glutathione reductase (GR) activity; one unit of enzyme activity is defined as 1 μmol of GS-TNB formed min^−1^ due to reduction of DTNB. (**i**) Percent relative water content (RWC). Values are mean ± SE (*n* = 3). Different letters on the top of bars indicate significant differences at *p* ≤ 0.05 level as determined by Duncan’s multiple range test (DMRT).

**Figure 5 antioxidants-11-00770-f005:**
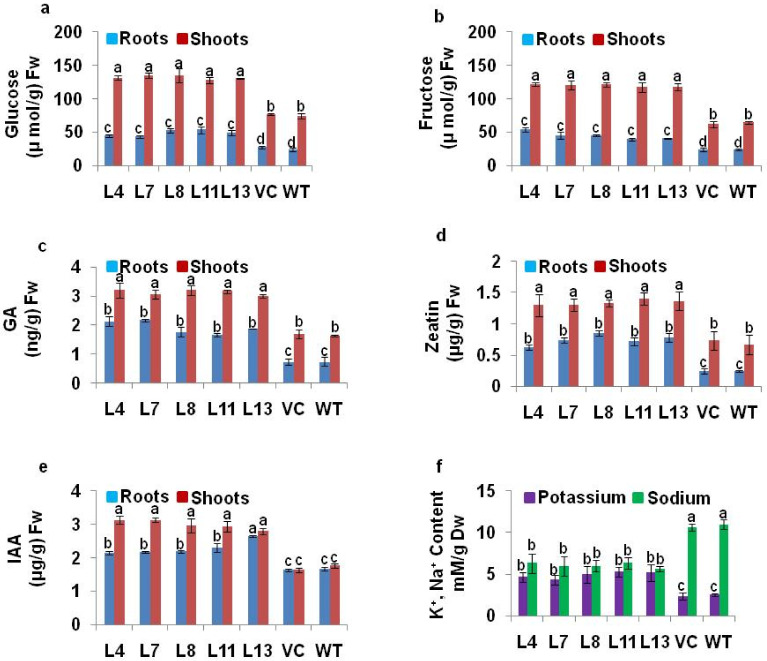
Soluble sugar, hormones, and K^+^ and Na^+^ content in the roots and shoots of *PDH45*-overexpressing marker-free transgenic lines (L4, L7, L8, L11, L13) as compared to WT and VC plants exposed to 24 h at 200 mM NaCl treatment. (**a**) Glucose content. (**b**) Fructose content. (**c**) Endogenous GA content. (**d**) Endogenous zeatin content. (**e**) Endogenous IAA content. (**f**) Endogenous potassium and sodium content. Values are mean ± SE (*n* = 3). Different letters on the top of bars indicate significant differences at *p* ≤ 0.05 level as determined by Duncan’s multiple range test (DMRT).

## Data Availability

The data presented in this study are available in the article.

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
