# Peer review of "Marker-Free Rice (Oryza sativa L. cv. IR 64) Overexpressing PDH45 Gene Confers Salinity Tolerance by Maintaining Photosynthesis and Antioxidant Machinery"

_antioxidants, 2022, doi:10.3390/antiox11040770_

Round 1
Reviewer 1 Report
Please see attached file for details

Author Response
Major issues
The theme of the article is scientifically highly relevant.
Q: Since soil has been used to compare the treatments, Does this treatment influence the microbial diversity in the rhizosphere with changes n salinity levels?
Response: Reconstituted soil was used throughout the experiment to avoid the effect of any other factor for all the treatments. Our previous research has laso shown that there is no impact on microbial community. Please refer to
Sahoo RK, Tuteja N. (2013) Effect of salinity tolerant PDH45 transgenic rice on physicochemical properties, enzymatic activities and microbial communities of rhizosphere soils. Plant Signaling & Behavior 8:8, e24950.
Q: As the theme is well established, the literature published prior to 2005 may be reduced to the minimum.
Response: As suggested by the reviewer, we checked the literature cited prior to 2005 is only of methodology part or pioneer work, therefore kept and few changed with recent one.
Kenmore, P. Sustainable rice production, food security and enhanced livelihoods, in: Rice Science: Innovations and Impact for Livelihood, Beijing, China, 2003, pp. 27-34.
is replaced with
Bandumula, N., 2018. Rice Production in Asia: Key to Global Food Security. Proceed. Natl. Acad. Sci., India Sec. B: Biological Sci. 2018, 88, 1323–1328.
Puchta, H. Marker-free transgenic plants. Plant Cell Tissue Organ Cult. 2003, 74, 123–134.
is replaced with
Sarkar, S.; Roy, S.; Ghosh, S.K. Development of marker-free transgenic pigeon pea (Cajanus cajan) expressing a pod borer insecticidal protein. Sci. Rep. 2021, 11, 10543.
Gu, R.; Liu, Q.; Pie, D.; Jiang, X. Understanding saline and osmotic tolerance of Populus euphratica suspended cells. Plant Cell Tissue Organ Cult. 2004, 78, 261-265.
is replaced with
Pérez-Jiménez, M.; Olaya Pérez-Tornero, O. In Vitro Plant Evaluation Trial: Reliability Test of Salinity Assays in Citrus Plants. Plants 2020, 9, 1352; doi:10.3390/plants9101352
Toki. S. Rapid and efficient Agrobacterium-mediated transformation in rice. Plant Mol. Bio. Rep. 1997, 15, 16-22.
is replaced with
Xiang Z, Chen Y, Chen Y, Zhang L, Liu M, Mao D and Chen L (2022) Agrobacterium-Mediated High-Efficiency Genetic Transformation and Genome Editing of Chaling Common Wild Rice (Oryza rufipogon Griff.) Using Scutellum Tissue of Embryos in Mature Seeds. Front. Plant Sci. 13:849666. doi: 10.3389/fpls.2022.849666
Lines 44-45: Major abiotic stresses (salinity, drought, extreme temperatures, heavy metal etc.) are significant limitations in rice cultivation globally.
Corrected as suggested in the revised MS.
Q: The role of strigolactones may be highlighted. (See Ref) Ref.: Regulation of Plant Mineral Nutrition by Signal Molecules. Microorganisms 2021, 9(4), 774; https://doi.org/10.3390/microorganisms9040774
Lines 93- 94: “…predictable manner avoids the transfer of undesirable transgenic material to non-transgenic crops and related species. …”.
Response: Thanks for the reviewer concern; the present research reported the development of marker free transgenic rice for salinity stress tolerance. We are unable to find a suitable place to fit the role of strigolactones in the MS.
Q: Since the host and transgenics differ only by a limited amount of foreign DNA material, how can it influence / increase the chances of transfer of undesirable material?
Line 191: 3.1. Molecular analysis of marker-free PDH45 transgenic lines
Response: As plasmid T-DNA contains some antibiotic marker gene (e.g. Hygromycin) along with the gene of interest, so it may transfer to the host cell. So here undesirable material refers to antibiotic marker genes.
Q: Has the strategy been tested in transgenic lines of rice available to other scientists? Can this strategy be applied to wheat, maize, or related cop plans?
Response: PDH45 has a proven track record to confer salinity tolerance in different crop and model plants and here worked well with IR64. With the same marker-free gene construct (pCAMBIA1300-PDH45), it can be tested on other economically important crops. The strategy has not been tested in transgenic lines of rice available to other scientists. It can be done in a future course of time.
Minor issues
Line 35: “which can potentially avoid public and biosafety concerns and facilitates the commercialization of genet ..” Maybe revised as: “which can potentially avoid public and biosafety concerns and facilitate the commercialization of genet ..”
Response: As suggested by the reviewer, the sentence has been modified in the revised MS.

Reviewer 2 Report
The manuscript describes the acquisition of tolerance to salinity stress by overexpression of PDH45 in rice.
The manuscript devised a method to select transformants based on salinity stress tolerance rather than antibiotic resistance. The obtained transformants clearly show tolerance to osmotic stress and maintenance of photosynthetic activity, and an increase in antioxidant activity. However, the following points need to be corrected or addressed before accepting the manuscript.
- Regarding selecting transformants using salinity stress tolerance, it is necessary to explain why 200 mM NaCl was used. It would be good to show data on whether the percentage of transformants or non-transformed escapes that appear depends on NaCl concentration. Also, has the selection with 200 mM NaCl caused any mutations in the rice genome? Placing cultured cells in a stress environment may lead to the accumulation of mutations that can impart tolerance to them. If a selection medium of 200 mM NaCl eliminated all non-transformants or VC lines, this would indicate that the phenotype of the resulting transformants is due to overexpression of PDH45.
- Regarding the quantification of gibberellin, how is GA3 identified? Reference 42 cited in the “Materials and methods” section uses ELISA and should not be able to distinguish between GA3 and GA1. Also, GA1 content is higher in rice than GA3, but the amount is on the order of ng/gFW. This differs from the GA3 content in the manuscript. Information on the gibberellin content of rice, for example, can be found in the following literature.
Tomoaki Sakamoto, Koutarou Miura, Hironori Itoh, Tomoko Tatsumi, Miyako Ueguchi-Tanaka, Kanako Ishiyama, Masatomo Kobayashi, Ganesh K. Agrawal, Shin Takeda, Kiyomi Abe, Akio Miyao, Hirohiko Hirochika, Hidemi Kitano, Motoyuki Ashikari, Makoto Matsuoka, An Overview of Gibberellin Metabolism Enzyme Genes and Their Related Mutants in Rice , Plant Physiology, Volume 134, Issue 4, April 2004, Pages 1642?1653, https://doi.org/10.1104/pp.103.033696
Hiroshi Magorne, Takahito Nomura, Atsushi Hanada, Noriko Takeda-Kamiya, Toshiyuki Ohnishi, Yuko Shinma, Takumi Katsumata, Hiroshi Kawaide, Yuji Kamiya and Shinjiro Yamaguchi
CYP714B1 and CYP714B2 encode gibberellin 13-oxidases that reduce gibberellin activity in rice
https://www.pnas.org/doi/10.1073/pnas.1215788110
Author Response
Comments and Suggestions for Authors
The manuscript describes the acquisition of tolerance to salinity stress by overexpression of PDH45 in rice.
The manuscript devised a method to select transformants based on salinity stress tolerance rather than antibiotic resistance. The obtained transformants clearly show tolerance to osmotic stress and maintenance of photosynthetic activity, and an increase in antioxidant activity. However, the following points need to be corrected or addressed before accepting the manuscript.
Regarding selecting transformants using salinity stress tolerance, it is necessary to explain why 200 mM NaCl was used. It would be good to show data on whether the percentage of transformants or non-transformed escapes that appear depends on NaCl concentration. Also, has the selection with 200 mM NaCl caused any mutations in the rice genome? Placing cultured cells in a stress environment may lead to the accumulation of mutations that can impart tolerance to them. If a selection medium of 200 mM NaCl eliminated all non-transformants or VC lines, this would indicate that the phenotype of the resulting transformants is due to overexpression of PDH45.
Response: The previous research of our group reported that PDH45 provides salinity tolerance in other rice varieties like PB1 and other model plants, therefore, 200 mM NaCl was chosen for the screening of transformants containing PDH45 gene. Further, in one of our previous reports (PNAS; Sanan-Mishra et al 2005), we have observed through qRT PCR, that the transcript level increases after exposure to 200 mM NaCl stress. So here we choose this concentration. The selection medium containing 200 mM NaCl eliminated all the non-transformants. The screened transgenic line doesn’t show any abnormal phenotype, therefore, it may be assumed that there is no mutation in the rice genome. The objective of the present study was to develop a marker free approach to avoid the use of antibiotic/herbicide tolerance genes and yes the phenotype of the transformants is due to overexpression of PDH45.
Regarding the quantification of gibberellin, how is GA3 identified? Reference 42 cited in the “Materials and methods” section uses ELISA and should not be able to distinguish between GA3 and GA1. Also, GA1 content is higher in rice than GA3, but the amount is on the order of ng/gFW. This differs from the GA3 content in the manuscript. Information on the gibberellin content of rice, for example, can be found in the following literature.
Tomoaki Sakamoto, Koutarou Miura, Hironori Itoh, Tomoko Tatsumi, Miyako Ueguchi-Tanaka, Kanako Ishiyama, Masatomo Kobayashi, Ganesh K. Agrawal, Shin Takeda, Kiyomi Abe, Akio Miyao, Hirohiko Hirochika, Hidemi Kitano, Motoyuki Ashikari, Makoto Matsuoka, An Overview of Gibberellin Metabolism Enzyme Genes and Their Related Mutants in Rice , Plant Physiology, Volume 134, Issue 4, April 2004, Pages 1642?1653, https://doi.org/10.1104/pp.103.033696
Hiroshi Magorne, Takahito Nomura, Atsushi Hanada, Noriko Takeda-Kamiya, Toshiyuki Ohnishi, Yuko Shinma, Takumi Katsumata, Hiroshi Kawaide, Yuji Kamiya and Shinjiro Yamaguchi CYP714B1 and CYP714B2 encode gibberellin 13-oxidases that reduce gibberellin activity in rice
https://www.pnas.org/doi/10.1073/pnas.1215788110
Response: Thanks for the concern of the reviewer to point out the typo error regarding GA content. Marker free transgenic rice plants were successfully raised using NaCl screening method and many physiological and biochemical attributes including hormones were tested to establish salinity tolerance in marker free transgenic lines.
We agree to the remark of the reviewer regarding the GA content and level difference of GA1 and GA3, the typo error has been corrected in the revised MS as GA content (ng/g) Fw.
Round 2
Reviewer 2 Report
The revised manuscript has addressed the issues raised in the previous review. The manuscript is now acceptable for publication.